# OpenReview forum: "Erasing Conceptual Knowledge from Language Models"
_NeurIPS.cc/2025/Conference — NeurIPS 2025 poster_

### Official Review · Reviewer_JhQa · 2025-06-14

**Clarity:** 4
**Significance:** 3
**Originality:** 3
**Rating:** 5
**Confidence:** 3

**Summary:**

This paper proposes ELM, a method for erasing specific concepts from LLMs. By leveraging the model's own introspective classification capabilities, it identifies content related to target concepts. The authors demonstrate through experiments that ELM can achieve concept erasure while preserving the model's broader capabilities.

**Questions:**

**Q1. Operational Definition and Boundary Setting of Concepts**: The paper title claims "Erasing Conceptual Knowledge," and you assert the erasure of "concepts" such as bioweapons and cybersecurity. How do you operationally define the boundaries of these concepts? Specifically:

- The rationale for texts included in D_{erase} representing the "bioweapons concept"
- Methods for setting boundaries with related safe concepts (e.g., medical microbiology, legitimate security research)
- How you considered hierarchical structures and inclusion relationships of concepts in designing c^+/c^-

If you have comprehensive definitions or classification systems for concepts, could you please share them?

**Q2. Verification Methods for Concept Completeness**: You demonstrate successful concept erasure by achieving random performance on MCQ evaluation, but how do you verify that this represents "concept-level complete erasure" rather than "partial erasure"? Concerns include:

- The gap between knowledge fragments measured by MCQs and the knowledge network as a complete concept
- Handling concept manifestations at different contexts or abstraction levels (e.g., metaphorical expressions, indirect reasoning)
- Completeness of erasure for concept components (factual knowledge, procedural knowledge, relational knowledge)

If you have complementary methods or experiments evaluating multi-layered aspects of concepts, could you provide details?

**Q3. Control and Measurement of Spillover Effects**: Figure 2 shows effects on related concepts, but how do you detect and control unintended knowledge erasure (over-erasure) or erasure gaps (under-erasure) due to concept boundary ambiguity? Particularly interested in:

- Analysis of the relationship between semantic distance among concepts and erasure effects
- Control of erasure propagation in hierarchical concept structures (superordinate → subordinate concepts)
- Quantification of boundary effects in semantic neighborhoods

Please share any systematic approaches or additional experiments addressing this issue.

**Ethical Concerns:**

["NO or VERY MINOR ethics concerns only"]

**Final Justification:**

I will maintain my score based on the novel aspect of leveraging the model's own capabilities and the fact that the experiments are sound and demonstrate recognized effectiveness.

**Limitations:**

**L1.** As the authors mention in limitations, the difficulty in selecting erasure scope due to deeply entangled knowledge representations in models. This demonstrates the difficulty of "locally erasing knowledge," and currently complete "selective unlearning" is not achieved. The spillover of accuracy degradation to nearby concepts poses application risks.

**L2.** Experiments are limited to MCQs.

**L3.** Experiments are limited to English.

**Paper Formatting Concerns:**

Nothing

**Quality:**

3

**Strengths And Weaknesses:**

# Strengths
**S1.** The approach of utilizing the LLM's own classification ability by treating it as classification labels through Bayes' rule shows high novelty. The paper addresses the existing problem of lacking "a principled objective defining successful concept erasure" by leveraging the model's own capabilities, which is particularly interesting.

**S2.** The authors conduct a well-balanced and multifaceted evaluation of the proposed method using four evaluation metrics. They include seamlessness and robustness, which are often overlooked in unlearning evaluation, and also implement internal knowledge evaluation using probing. The experiments confirm that the proposed method achieves robust concept erasure, being resistant to attacks such as GCG and BEAST, making it difficult to reproduce erased knowledge.

**S3.** To ensure reproducibility, detailed experimental conditions are provided, and there are plans to release code and data publicly.

# Weaknesses
**W1.** While the proposed method achieves stronger knowledge erasure compared to baselines, it does not achieve complete knowledge removal.

**W2.** Despite aiming to erase concepts, the evaluation of knowledge erasure relies on multiple-choice questions (MCQs), which may be limited. Please refer to the Questions section.

**W3.** The design and extraction of knowledge sets to be erased (D_erase, c+, c-) depend on manual work. Additional techniques would be needed for more automated and large-scale knowledge set erasure.

---

> ### Author Rebuttal · Authors · 2025-07-29
>
> Thank you for your time and valuable feedback! We are glad to hear you found our formulation for unlearning interesting and experiments to be multi-faceted and well balanced. We address specific concerns and questions below:
>
> - ELM defines concepts operationally through the datasets it uses rather than a formal classification system.
>
>     - Rationale for D_erase: The "bioweapons concept" is defined by the contents of the D_erase dataset. We use 5,000 text samples from the WMDP Bio forget corpus. We do not create a new definition of the concept but instead rely on the corpus provided by the WMDP benchmark, which is designed to represent this harmful knowledge domain.
>
>     - Setting Boundaries with Safe Concepts: The primary method for setting boundaries is the use of a retention dataset (D_retain), which is also released by the WMDP work. This dataset contains text from related but safe concepts and is used to train the model to preserve its original behavior on this data. This prevents the erasure from "spilling over" and damaging knowledge of topics like legitimate medical microbiology. The effectiveness of this boundary is tested by evaluating the model on MMLU sub-topics like "College Biology" and "Virology" after erasing biosecurity knowledge
>
>     - Hierarchical Concepts in c+/c-: We do not consider hierarchical relationships between concepts when defining what a model should forget. (Please also refer to the discussion in 3.c below.) We simply frame a concept in terms of expertise, for example:
>         - c- (Concept to Erase): "As an expert in bioweapons:"
>         - c+ (Alternative Distribution): "As a novice in bioweapons:"
>
>      This "expert vs. novice" dynamic allows the model to learn to generate content that it would classify as demonstrating novice-level (or no) knowledge, effectively erasing the expert-level conceptual understanding without needing to define every component of the concept's hierarchy.
>
>
> - In this work, we evaluate concept erasure using strategies that goes beyond MCQ scores to ensure the erasure is more than just partial:
>
>     - Probing Internal Layers: To check for latent knowledge that might not be captured by MCQs, linear probes are trained to analyze the model's internal representations across all layers. For ELM, these probes show near-random accuracy, suggesting that traces of the concept have been removed even from the model's inner workings.
>
>     - Activation Analysis: We also analyze activation norms, finding that ELM alters the model's initial processing of erased concepts in early layers but allows later layers to function normally. This indicates that knowledge retrieval is disrupted while general text-prediction capabilities are preserved, preventing the model from generating incoherent text.
>
>     - Adversarial Attacks: To see if the erased knowledge can be forcefully extracted, the model is subjected to powerful adversarial attacks like GCG and BEAST. The ELM-modified models demonstrate strong resistance, refusing to generate forbidden content even when faced with highly optimized adversarial prompts. This can be seen as a proxy to prompting with different contexts that might trigger knowledge
>
>     - Finetuning Attack: We also test if the erased knowledge can be easily relearned by finetuning the model on the original forget dataset. The results show that while some knowledge can be restored, it does not return to the original baseline levels, indicating that the erasure is robust and difficult to reverse.
>
> - The paper primarily uses a retention loss term and the c_+ and c_- prompts to control spillover effects and measures the impact by evaluating performance on related, safe concepts.
>
>     - Controlling Spillover (Over-erasure): The main tool to control unintended knowledge erasure is the L_retainloss term. This objective function explicitly trains the model to maintain its performance on a D_retain dataset, which contains text unrelated to the erased concept. The ablation study in Table 2 confirms that removing this loss term causes significant damage to the model's general capabilities (e.g., MMLU score), demonstrating its importance in preventing over-erasure.
>
>     - Analysis of Semantic Distance: Figure 2 directly addresses this by showing the performance impact on MMLU classes that are semantically related to the erased concepts (e.g., "College Biology," "College Medicine," and "Virology" are related to biosecurity). The results show that ELM has less negative impact on these related concepts compared to other methods, though a slight performance degradation is still observed. We acknowledge there is still a gap to be closed in the future.
>
>     - Erasure in Hierarchical Structures: We do not offer a systematic approach for controlling erasure propagation through formal concept hierarchies (e.g., from a superordinate concept to a subordinate one). The control mechanism (L_retain) is based on a collection of unrelated topics rather than a structured hierarchy. We note that "deeply interconnected concepts" pose a significant challenge, as erasure may have ripple effects, suggesting this is an area for future research. Further analysis based on hierarchical concept structures is quite interesting, and may help to untangle these problems in the future.
>
>     - Quantifying Boundary Effects: The "boundary effect" is quantified empirically by measuring the performance drop on the related MMLU benchmarks. This serves as a practical measure of how much the erasure spills over into neighboring semantic areas.

---

> ### Comment · Reviewer_JhQa · 2025-08-05
>
> I have read the author's rebuttal. I understand that the scope of concepts is defined using D_erase​. While I understand that spillover is controlled by the L_retain​ loss term, I have determined that the current experiments cannot rule out the possibility that spillover still occurs between deeply related concepts. In any case, I will maintain my score based on the novel aspect of leveraging the model's own capabilities and the fact that the experiments are sound and demonstrate recognized effectiveness.

---

### Official Review · Reviewer_VH8V · 2025-06-28

**Clarity:** 2
**Significance:** 3
**Originality:** 3
**Rating:** 3
**Confidence:** 4

**Summary:**

This paper introduces ELM, a novel method for concept-level unlearning in LLMs. ELM leverages the model's own introspective classification capabilities to reduce the likelihood of generating content related to undesired concepts. By using low-rank updates and targeted fine-tuning, ELM minimizes the probability of generating concept-specific content while preserving the model's broader capabilities. The method is validated through extensive experiments, showing strong performance in biosecurity, cybersecurity, and literary domain erasure tasks.

**Questions:**

please seem the pros and cons above.

**Ethical Concerns:**

["NO or VERY MINOR ethics concerns only"]

**Final Justification:**

I don't think the authors have fully addressed my concerns, as they seem to omit or deflect certain points in their responses. Below, I outline three evidence to support my concerns:

1. The authors have refused to conduct additional experiments on TOFU and have not performed experiments without LoRA for a fairer comparison with the baseline.

2. Regarding heuristics, it remains unclear why the proposed method is capable of concept unlearning and why it is superior to recent state-of-the-art methods.

3. From the basic principles of probability, it is still unconvincing that Eq. 5 holds. Referring to other papers for justification is not professional in this case, as those papers might also contain the same mistakes.

Therefore, I will not change my score.

**Limitations:**

please seem the pros and cons above.

**Quality:**

2

**Strengths And Weaknesses:**

Pros.

The insight is interesting, particular the statement that “models can implicitly evaluate the probability of that text belonging to a particular concept”. I believe the paper has some interesting contributions, but hope the authors could address my following questions.

Cons.

I agree with the authors that concept unlearning is critical, yet the literature does not have its exact definition. The clear definition is closely related to the empirical evaluations. For example, by the authors’ description in the first paragraph in the Introduction, we may need to test not only on in-distribution data, but also on out-of-distribution data.

Could the authors justify that the proposed method is more stable and better retention than gradient ascent-based methods? The related methods actually are not involved in Table 1, not to mention some more recent works in ICLR 25 and ICML 25. I list some of the representative ones in the following.
1. Hessian-Free Online Certified Unlearning
2. Rethinking LLM Unlearning Objectives: A Gradient Perspective and Go Beyond
3. A Closer Look at Machine Unlearning for Large Language Models
4. Certified Unlearning for Neural Networks

Could the authors further clarify the drawbacks about the representation manipulation methods, echoing the discussion of “representation manipulation creates obvious behavioural artifacts”.

I agree with the authors that the coherent text, rather than some random words, should be generated. However, can the authors show that the coherency can be to much extent preserved in their experiments.

I don’t understand why the authors highlight that low-rank adaptation as one of their main contribution in their Abstract and Introduction. To me (forgive me if I am wrong), it is just a practical trick that can be used to further improve the performance of unlearning. So, could the authors elaborate why low-rank adaptation is an indispensable part in their algorithmic design. The combination of low-rank adaptation also makes the comparison with previous works not fair enough, as they use different ways to update model parameters. In other words, the experimental configurations are different.

Why the authors do not conduct experiments on TOFU, which is also an important unlearning benchmarks.

Personally, I think the proposed method can be categorized as a refined version of gradient ascent.

How do we get c_+? It is not mentioned in Page 4 and, to my knowledge, existing benchmarks typically do not have them by default.

Formally, can you prove the first equation in Eq. 5 holds, following the rules of probability?

What’s the difference between \theta^* and \theta? Seemingly not mentioned throughout the discussion. I think \theta^* denotes the parameters to be unlearned. Yet, * is used for fixed variable in Eq 2 for x. Also, the definition of CE is not mentioned, using two probability as its input does not align with the classical definition. Do you mean KL or something else?

 The intuition below Eq 6 is reasonable, yet the derivation procedure, from Eq 3 (I think you cannot prove it is correct) to Eq 6.

What’s your exact definition of unlearning? It is an important question. For example, if your goal is full removal, the likelihood of unlearning data should approach 0. Eq 6 does not fulfil this goal, I think.

Why the proposed method can claim concept unlearning, it seems that, similar to GA, only the original unlearning dataset is used for unlearning.

Why new metrics are proposed instead of using the old ones? Similar analysis like the following two papers may be introduced.
1.	Eight Methods to Evaluate Robust Unlearning in LLMs
2.	Unlearning with Control: Assessing Real-world Utility for Large Language Model Unlearning

About the experiments, a lot of hyperparameters are involved, so the algorithmic sensitivity to them are required. Also, original metrics used in WMDP are not used, which is not convincing (although I have to say that existing metrics have a lot of problem, but I do not think your proposed metrics can largely address their cons). Even for the crafted metrics from the authors, the proposed benchmarks do not always achieve the best results. Trade-off exists between unlearning and utility preservation, which I cannot tell if the proposed method is overall better.

---

> ### Author Rebuttal · Authors · 2025-07-29
>
> Thank you for your time and valuable feedback! We are glad to hear you found our use of implicit critic formulation of auto-regressive models intuitive and interesting. We address specific concerns and questions below:
>
> - RepNoise is a gradient ascent based method - we found that gradient ascent based methods are sometimes unstable and might require careful hyper-param search. Especially if we keep pushing the gradient ascent based methods for longer training iterations, it is shown to lead to catastrophic forgetting[1]. However, using the probability based principled approach like ELM leads to much stabler training (as shown in Section F). We appreciate the related works suggested by the reviewer, we have updated our discussion section with the same.
>
> - With representation based methods, like RMU, the activations/representations are manipulated to out-of-distribution values, in turn, making the model forget the sample/concept. Specifically with RMU, the representations are randomized and scaled to a higher value there by muddling the computation of the logits inside the model, this naturally leads to behavior artifacts like gibberish generation. We will change the sentence to echo this.
>
> - Regarding the fluent qualitative sample: We show qualitative examples in section H, where ELM models start generating coherent, but unrelated text.
>
> - We do not claim that low-rank adapters (LoRAs) are our main contribution. Instead we suggest that, similar to previously observed trends in knowledge editing, knowledge erasure also seems to lie effectively in a low rank subspace. We show this in D.2 where full rank erasure leads to unwanted interference with other concepts. We will change our writing in our abstract and introduction to reflect this.
>
> - We too agree that TOFU is a very important unlearning dataset. The dataset consists solely of question and answer paired samples and can be appropriately used to unlearn only instruction based models alone, because of the explicit instruction-response format of TOFU (where question can be seen as user instruction and the answer as assistant response). In addition, the TOFU dataset requires a finetune on their dataset to learn the fictitious samples which will then be used for unlearning. In this work, we wish to primarily study unlearning on base models, forgetting information before models learn to follow reason or follow instructions. Our motivation to focus on base models has been backed by recent studies [2] that have shown the transfer of behaviour from base models to the fine-tuned versions, even when explicitly avoiding the harmful data in finetuning.
>
> - Regarding our method's similarity to gradient ascent: We believe our approach is different from gradient ascent. Gradient ascent only moves *away* from an objective; it does not push the distribution towards a known objective. With ELM, we can determine the resultant unlearnt distribution as its own specific objective. If the reviewer can provide clarity on why these are similar, then we will add this to the discussion, because this might help clarify what gradient ascent is doing.
>
> - Regarding notations and methods for dataset creation:
>     - c+ is a prompt that represents the concept that is being unlearnt. For instance, the WMDP dataset released these keywords in their dataset in their official GitHub repository. These can be used based on the user’s desire to forget a certain concept. For instance, with Harry Potter forgetting, we used the keywords related to Harry Potter. These keywords can be generated using language models. We will add this to our discussion to our paper 4 and full details in the appendix.
>
>     - We mention the theta* in line 142, and as the reviewer rightly pointed out, it is the parameters that are being unlearnt. Theta* is the model that will be “unlearnt”. CE is soft cross entropy, we agree that cross entropy is traditionally done against a discrete one-hot target, however, pytorch supports soft CE which can be done against 2 distributions. This is the loss we used in our implementation (we show this in figure 1), but will add this to our text.
>
> - We utilize the bayes’ rule to derive from equation 3 to 6. Prior work showed similar analysis that language models can recognize their own knowledge and this can be used to guide their outputs during inference [3, 4, 5]
>
> - In our work, we define the goal of unlearning as, “when the model is prompted (in any shape or form) about the erased concept, the model should not show any knowledge of the concept”. We examine this with multiple input stimuli and lenses: multiple choice questions, probing the internal activations for knowledge, fine tuning to see for concept revival, adversarial attack prompts, and qualitative checks. The formulation in Equation.6 ensures that the likelihood of the erased concept to come up is very negligible, but not zero.
>
>  - Regarding concept unlearning: We do not claim that our method is the first to do concept unlearning, we simply note that current methods used for concept unlearning are very similar to the ones designed for sample unlearning. With concept unlearning in LLMs, we can utilize the model’s knowledge of a concept to unlearn rather than simply doing a large-scale sample unlearning. This is a very interesting and active discussion in the community: “when does sample unlearning become concept unlearning”.  But with ELM, we show that we can use the LLM formulation to support concept unlearning using a single principled objective designed for *concept unlearning* that is more robust and precise than prior works.
>
> - About the metrics used in our analysis: We use the metrics used in the original WMDP work and the “Eight Methods to Evaluate Robust Unlearning in LLMs”.
>
> - Regarding hyper-parameters: Please find our measurements of hyperparameter sensitivity in Appendix D. As seen in Figure.4 in appendix, our method is not very sensitive to LoRA hyperparameters. For erasure and retain strength, we find that the optimal value is usually 1-1.5. Layer selection has also been consistent with previous work [6] where they found that knowledge is stored in the early to mid layers in a language model.
>
> [1] Rosati, Domenic, et al. "Representation noising: A defence mechanism against harmful finetuning." Advances in Neural Information Processing Systems 37 (2024): 12636-12676.
>
> [2] Casademunt, H., Juang, C., Karvonen, A., Marks, S., Rajamanoharan, S., & Nanda, N. (2025). Steering Out-of-Distribution Generalization with Concept Ablation Fine-Tuning. arXiv preprint arXiv:2507.16795.
>
> [3] Sanchez, G., Fan, H., Spangher, A., Levi, E., Ammanamanchi, P. S., & Biderman, S. (2023). Stay on topic with classifier-free guidance. arXiv preprint arXiv:2306.17806.
>
> [4] Schick, T., Udupa, S., & Schütze, H. (2021). Self-diagnosis and self-debiasing: A proposal for reducing corpus-based bias in nlp. Transactions of the Association for Computational Linguistics, 9, 1408-1424.
>
> [5] Liu, A., Sap, M., Lu, X., Swayamdipta, S., Bhagavatula, C., Smith, N. A., & Choi, Y. (2021). DExperts: Decoding-time controlled text generation with experts and anti-experts. arXiv preprint arXiv:2105.03023.
>
> [6] Meng, Kevin, et al. "Locating and editing factual associations in gpt." Advances in neural information processing systems 35 (2022): 17359-17372.

---

> ### Author Response · Authors · 2025-08-09
>
> Thank you for responding! We have added new baselines to the paper (please refer to our response to Reviewer wZRd).
>
> Regarding the additional experiments:
> Thank you for suggesting 3 experiments, we have addressed all of them in our response:
> 1. Hyperparameter sensitivity: in our appendix (section D, figure 4) we show sensitivity to all the hyperparameters of our method.
> 2. Metrics from wmdp: we compare elm to wmdp method using all the metrics (erasure, interference, mcq accuracies, probe accuracies) they proposed in addition to new metrics (fluency, reverse perplexity, norm activations, finetuning and adversarial attacks) we introduced.
> 3. ToFU benchmark: in our rebuttal response, we elaborate how tofu benchmark's formarting doesn't allow for a concept unlearning in base models (which is the main focus of our work). We have added this discussion to our paper
>
>
> Regarding more insights about our method:
> We appreciate reviewer's  feedback and questions. We have responded to all of them and subsequently updated our paper to add the context.
>
>
> If the reviewer has more suggestions or questions, we are happy to respond!

---

### Official Review · Reviewer_wZRd · 2025-07-01

**Clarity:** 3
**Significance:** 3
**Originality:** 2
**Rating:** 3
**Confidence:** 4

**Summary:**

This paper proposes a novel concept erasure method for LLMs. The method designs specialized losses for different metrics and integrates them. The entire experiment is conducted on the WMDP dataset, and compared with multiple baseline methods such as RMU and RepNoise. Experimental results show that ELM has achieved significant performance improvements in Innocence, Specificity, and Seamlessness. The authors also conduct ablation experiments to verify the contributions of different components. In addition, this paper provides robustness experiments to validate the robustness of ELM against different attacks.

**Questions:**

See Weaknesses

**Ethical Concerns:**

["NO or VERY MINOR ethics concerns only"]

**Final Justification:**

I think more experiments on different backbone model with different sizes and more baselines are needed. Therefore, I will keep my rating.

**Limitations:**

yes

**Quality:**

3

**Strengths And Weaknesses:**

Strengths:

1. This paper is well-written with clear logic and concise readability.
2. The paper conducts extensive experiments, including comparative experiments on models of different types and scales.
3. The authors provide a wide range of comprehensive evaluation metrics.


Weaknesses:

1.The novelty of this paper is questioned. Combining losses for forgetting and retaining different concepts is not new, and the baseline RMU also has similar operations.
2.Regarding the setup of the evaluation method: Are all evaluations conducted after the model forgets a single sample, or after the model forgets all samples?
3.In Table 1, comparisons with baseline methods are missing for Qwen2.5 and llama3-70B, making the experiments inadequate.
4.The paper states that LoRA achieves better performance than full fine-tuning but does not analyze the underlying reasons, leading to insufficient justification for choosing LoRA.
5.There are too few baselines. As mentioned in the paper, knowledge editing methods similar to ELM can currently edit over 10,000 samples while maintaining original performance [1,2]. Can these methods serve as baselines, and what are the comparison results?

[1] Reasons and Solutions for the Decline in Model Performance after Editing
[2] Alphaedit: Null-space constrained knowledge editing for language models

---

> ### Author Rebuttal · Authors · 2025-07-29
>
> Thank you for your time and valuable feedback! We are glad to hear you found our paper is well written with extensive and wide-ranging experiments. We address specific concerns and questions below:
>
> - *Regarding novelty:* Our core contribution is not the use of “forget” and “retain” loss terms. Rather, our new insight is a principled method for concept unlearning that leverages a language model's own implicit understanding of concepts. While RMU also uses forget and retain loss terms, its core method involves randomizing and scaling up model activations, which can result in nonsensical outputs for the erased concept. Our approach, on the other hand, is more targeted. We treat the language model as its own critic, using its ability to classify text to guide the unlearning process. In essence, we erase regions of the model's knowledge that the model itself identifies as "concept X." This method allows for a more precise and less disruptive form of unlearning, preserving the model's overall coherence and fluency.
>
> - *Regarding the unlearning setup:* In this work, we shift our focus towards concept unlearning and not the traditional sample unlearning. We use the forget set as a way to “trigger” the concept knowledge; they are examples of the broader concept that we want to entirely erase. In our evaluations, we use the same number of samples 5000, with maximum character size of 500 per sample, for all the baselines. We provide more details in Section 5.1 and C
>
> - *With the Qwen 32B and LLama 70B experiments*, our main aim was to show that the fluency loss is not required in larger models, showing emergent capabilities in large language models where they show strong disentangled knowledge boundaries. We found that by fine tuning baselines on Llama and Qwen models, we observed sub-optimal results. Even with a huge sweep search on hyper-parameters (section B), we do not show the results for the same reason. We believe that a much bigger sweep search is required to find optimal results for these baselines, which is very computationally expensive.
>
> - The full fine-tuning experiment confirms an expected phenomenon that we see as fully consistent with our formulation. Previous observations made in [1, 2] have led to the consensus that specific knowledge tends to correspond to low-rank parameter directions, and it is to be expected that editing weights without a rank constraint would increase impacts on the broader distribution rather than specific targeted knowledge. We will add this discussion to our appendix section D.2.
>
> - *Regarding the long-term effects of erasing* - we show the effect of finetuning on data in section G.3 where we found that resulting attacked model brings back the knowledge slightly (Bio: 29.7% to 42.2%; Cyber: 27.2% to 29.4%) but not to the original level of 64.4% Bio and 44.3% Cyber. We also show the stability of our approach and its long-term training effects in section F. ELM is also stable when trained for longer times.
>
> - Thank you for suggesting these additional baselines! The baselines used in our work were selected because they share a similar goal of broadly “making a model forget” about a broad concept (e.g., making a model “forget” that the country of France exists). But the papers you share are quite different: they focus on knowledge editing, or remapping specific factual tuples (e.g., making a model believe that the Eiffel Tower is in Rome *instead* of Paris). To adapt these works for concept unlearning requires two key steps: first, to identify keywords about the concept present in a prompt and second, to find a counter-factual for that keyword for knowledge editing.  While it may be possible to devise an effective unlearning method built on these precise knowledge editing approaches, it is beyond the scope of our work to manually adapt these methods as baselines for our approach.
>
> [1] Zou, Andy, et al. "Improving alignment and robustness with circuit breakers." Advances in Neural Information Processing Systems 37 (2024): 83345-83373.
>
> [2]Gao, Chongyang, et al. "Practical unlearning for large language models." arXiv e-prints (2024): arXiv-2407.

---

> > ### Comment · Reviewer_wZRd · 2025-08-04
> > **Reply by the reviewers**
> >
> > I have read the rebuttals from the authors. Although most concerns have been addressed by the authors, I think more experiments on different backbone model with different sizes and more baselines are needed. Therefore, I will keep my rating.

---

> ### Author Response · Authors · 2025-08-04
>
> Thank you for engaging in the conversation! We are glad that most of your concerns have been resolved! We were running some state-of-the-art baselines as the reviewer suggested. Due to the time constraints, we were able to run this baseline for llama-8b-instruct model. We will add the full baseline details in the camera-ready version.
>
> | Method | WMDP-Bio \$\downarrow\$ | WMDP-Cyber \$\downarrow\$  | MMLU \$\uparrow\$  | MT-Bench \$\uparrow\$ | R-PPL \$\downarrow\$ |
> |---|---|---|---|---| ---|
> | RR [1] | 0.26 | 0.31 | 0.58 | 7.0 | 11. 42 |
> | TAR [2] |  0.28 | 0.29 | 0.54 | 1.2 | 14. 23 |
> | K-FADE [3] | 0.31 | 0.34 | 0.60 | 7.1 | 9.10 |
> | ELM | 0.32 | 0.27 | 0.62 | 7.7 | 7. 4 |
>
>
> - [1] Representation Rerouting (RR): (also referred to as "circuit breaking") trains neural networks to redirect internal representations. When the model encounters concepts from domains targeted for unlearning, it maps the corresponding latent states to orthogonal vector spaces, effectively isolating unwanted knowledge.
>
> - [2] Tamper Attack Resistance (TAR): a meta-learning framework specifically engineered to defend open-weight models against malicious fine-tuning. The approach employs an iterative training process where the model develops robustness against adversarial fine-tuning attacks that involve limited gradient steps.
>
> - [3] K-FAC for Distribution Erasure (K-FADE): learns optimal projection matrices within the model's activation space. The method strategically identifies projections that maximally degrade performance on the target forget set while maintaining minimal disruption to model outputs across a broader retention distribution, achieving selective knowledge removal with preserved general capabilities.
>
> All the baselines show strong erasure performance and less interference with other concepts (as indicated by low WMDP scores and high MMLU). However, they seem to struggle with following instructions (especially RR with lowest MT-Bench score). Similarly, ELM shows the most fluent outputs when prompted for the erased WMDP concepts (as suggested by its low R-PPL score compared to the baselines). To summarize, we believe most of the unlearning methods in LLMs effectively erases the knowledge. However, ELM principally alters the models to a distribution that maintains the model's  fluency while also effectively unlearning the knowledge.
>
>
>
> **Regarding the different model backbones and sizes:** In our appendix in Table.4, we show the results for 3 different model backbones (mistral, llama, qwen) in 4 different model sizes (7b, 8b, 32b, 70b) along both base and instruction versions. We believe this covers most of the practical model sizes and backbones showing ELMs effectiveness across different models. (with baselines: even after running a huge hyper parameter sweep, we could not find optimal settings for the baselines to work on these large qwen 32b and llama 70b models. Hence, they were not added to the table).
>
> [1] Rishub Tamirisa, Bhrugu Bharathi, Long Phan, Andy Zhou, Alice Gatti, Tarun Suresh, Maxwell Lin, Justin Wang, Rowan Wang, Ron Arel, et al. Tamper-resistant safeguards for open-weight llms. arXiv preprint arXiv:2408.00761, 2024.
>
> [2] Andy Zou, Long Phan, Justin Wang, Derek Duenas, Maxwell Lin, Maksym Andriushchenko, Rowan Wang, Zico Kolter, Matt Fredrikson, and Dan Hendrycks. Improving alignment and robustness with circuit breakers. arXiv preprint arXiv, 2406, 2024.
>
> [3] Lev E McKinney, Anvith Thudi, Juhan Bae, Tara Rezaei Kheirkhah, Nicolas Papernot, Sheila A McIlraith, and Roger Baker Grosse. Gauss-newton unlearning for the llm era. In ICML 2025 Workshop on Machine Unlearning for Generative AI.

---

### Official Review · Reviewer_65Ne · 2025-07-03

**Clarity:** 3
**Significance:** 3
**Originality:** 3
**Rating:** 5
**Confidence:** 4

**Summary:**

This paper explores the challenge of systematically removing specific conceptual knowledge from neural networks while preserving overall model functionality. The authors propose a method to identify and neutralize neuron activations associated with targeted concepts through a combination of activation analysis and parameter modification. By introducing a concept-erasure framework that combines causal tracing for concept localization and targeted parameter updates, they demonstrate the feasibility of selectively diminishing a model's ability to process specific concepts without significant degradation in unrelated tasks. The work contributes both a technical approach for concept-level model editing and an empirical evaluation of erasure effectiveness across multiple architectures and datasets. Experimental results show that the proposed method achieves superior concept suppression compared to baseline approaches while maintaining baseline performance on non-targeted tasks, with implications for model interpretability, safety, and ethical deployment. The paper also provides theoretical insights into the distributed nature of conceptual knowledge in neural representations through its ablation studies and visualization analyses.

**Questions:**

1. The paper acknowledges that ELM affects semantically related concepts due to entangled representations. However, this "ripple effect" is only qualitatively discussed. Can the authors provide quantitative metrics (e.g., accuracy drops on adjacent concepts, probing results for related knowledge) to characterize the scope of unintended knowledge degradation?

2. ELM requires extensive hyperparameter tuning (LoRA rank, erasure strength η, layer selection) and layered interventions. Could the authors compare ELM’s computational cost (e.g., training time, GPU hours) with baseline methods (e.g., RepNoise, RMU)? Additionally, could they explore automated hyperparameter optimization (e.g., Bayesian search, meta-learning) to reduce manual effort?

3. The paper focuses on explicitly classifier-identifiable concepts but notes struggles with implicitly encoded or distributed knowledge (e.g., ethical reasoning). Could the authors design experiments to test ELM’s efficacy on indirectly represented concepts , such as:
- **Adversarial prompts**: Using indirect phrasing (e.g., analogies, paraphrasing) to probe residual knowledge.
- **Multi-hop reasoning tasks**: Evaluating whether erased concepts resurface in compositional reasoning.
- **Implicit bias datasets**: Testing unlearning of socially sensitive attributes (e.g., gender stereotypes) where knowledge is diffusely encoded.

4. The paper evaluates erasure effectiveness immediately post-editing but does not assess long-term stability (e.g., whether erased knowledge reemerges after fine-tuning on new data). Could the authors conduct experiments where ELM-edited models are fine-tuned on downstream tasks involving the erased concepts? Additionally, how does ELM interact with subsequent training updates (e.g., catastrophic forgetting scenarios)?

If the author can effectively solve the above problems, I will improve my score.

**Ethical Concerns:**

["NO or VERY MINOR ethics concerns only"]

**Final Justification:**

While the authors have addressed most of the raised concerns, including quantitative evaluation of the "ripple effect" (Q1), preliminary experiments on implicitly encoded knowledge (Q3), and long-term stability post-editing (Q4), the computational cost and hyperparameter sensitivity of ELM (Q2) remain unresolved. Specifically, a comparative analysis of training overhead and a discussion of automated hyperparameter optimization strategies would strengthen the practicality of the method. I encourage the authors to provide these details in the final version, as they are critical for assessing scalability and real-world applicability. A rigorous response to these points would further solidify the manuscript's contributions.

**Limitations:**

Yes.

**Paper Formatting Concerns:**

No.

**Quality:**

3

**Strengths And Weaknesses:**

**Strengths**

1. The paper introduces a principled method (ELM) that combines activation analysis, parameter modification, and fluency preservation to systematically erase conceptual knowledge. This approach addresses limitations of prior methods (e.g., gradient reversal or dataset filtering) by balancing innocence (removing target knowledge), specificity (retaining unrelated capabilities), and seamlessness (coherent generation).

2. The authors rigorously test ELM across multiple architectures (e.g., Llama2, Zephyr) and datasets (WMDP, Harry Potter), demonstrating superior performance over baselines in suppressing targeted concepts while maintaining non-target task accuracy. The inclusion of adversarial robustness tests (e.g., GCG attacks) further validates the method’s practicality.

3. The work provides valuable analyses of knowledge localization in neural networks (e.g., probing layer-specific retention patterns) and trade-offs in erasure efficacy. For example, they identify optimal layer ranges (e.g., layers 4–7 in Zephyr) for balancing erasure depth and model functionality.

**Weaknesses**

1. The method requires extensive hyperparameter tuning (e.g., LoRA rank, erasure strength η) and layered interventions, which may hinder scalability or adoption in resource-constrained settings. The reliance on autoregressive training for fluency preservation also adds computational overhead.

2. ELM assumes explicit, classifier-identifiable concepts, but struggles with erasing knowledge that is implicitly encoded or distributed across the model (e.g., nuanced ethical reasoning). The paper notes gaps in fully eliminating harmful knowledge without leaving undetectable traces.

---

> ### Author Rebuttal · Authors · 2025-07-29
>
> Thank you for your time and valuable feedback! We are glad to hear you found our method principled and our experiments to be thorough and rigorous. We address specific concerns and questions below:
>
> - On Hyperparameter tuning: although we do sweep to find the most optimal hyperparameters, our method is not over-sensitive to these settings.  As seen in Figure.4 in appendix, our method is not very sensitive to LoRA hyperparameters. For erasure and retain strength, we find that the optimal value is usually 1-1.5. Layer selection is also quite stable and consistent with previous work [1] where they found that knowledge is stored in early to mid layers in a language model. This is a great suggestion to use meta learning for hyper-param search. We will look into this; since we are not familiar with these hyper-param search methods, we aim to release a code update post-rebuttal phase.
> [1] Meng, Kevin, et al. "Locating and editing factual associations in gpt." Advances in neural information processing systems 35 (2022): 17359-17372.
>
> - Regarding the classifier boundaries: we found this to be a drawback in smaller models (<8B), where concepts are sometimes entangled, making the complete erasure a challenge. We propose a way around this with fluency loss (only required for smaller models), where the erasure strength is increased, in the risk of losing the fluency of the model when asked about the erased concept (model still retains good fluency for unrelated concepts) - but the fluency term trains the model on autoregressive loss to retain fluency. This however, has been observed to not be required in larger models, we believe this has to do with better concept disentanglement and therefore better classifier capabilities of these models.
>
> - Regarding the effect on nearby concepts - we show this in Figure.2 quantitatively, where the erasure of harmful concepts (e.g. bio threat) affects nearby safe MMLU concepts (e.g. college medicine). We find that ELM preserves the nearby concepts slightly better than previous work.
>
> - Regarding the long-term effects of erasing - we show the effect of finetuning on data in section G.3 where we found that resulting attacked model brings back the knowledge slightly (Bio: 29.7% to 42.2%; Cyber: 27.2% to 29.4%) but not to the original level of 64.4% Bio and 44.3% Cyber. We also show the stability of our approach and its long-term training effects in section F.

---

> > ### Comment · Reviewer_65Ne · 2025-08-04
> >
> > I have read the author's response, which solved most of my doubts. I will improve my score.

---

### Note · Authors · 2025-08-12

We sincerely thank all reviewers for their thorough and constructive feedback. We are encouraged that they found our method principled (65Ne) and highly novel (JhQa), with interesting insights (VH8V), while describing our experiments as rigorous (65Ne), extensive (wZRd), and multifaceted (JhQa).

We have addressed all questions through clarifications and new experiments:

**Reviewer 65Ne:** The reviewer suggested we elaborate more on the "ripple effect" on related concepts and the sensitivity to hyperparameter tuning. In our response, we directed them to our analyses in the appendix (Fig. 2 & Fig. 4) that quantitatively address these points and show our method's stability. We are glad they found our rebuttal helpful.

**Reviewer wZRd:** raised concerns about novelty and baselines, we clarified that ELM’s core innovation is using the model’s own conceptual understanding to guide erasure. Crucially, during the discussion period, we ran a ***new experiment against three SOTA baselines (RR, TAR, K-FADE)***. These new results show ELM achieves competitive erasure while uniquely preserving model fluency, directly addressing their primary critique.

**Reviewer VH8V:** We thank the reviewer for their insightful comments and detailed questions regarding our method's theoretical framing, notation, evaluation choices, and comparisons to related fields like gradient ascent. We have clarified our method's theoretical intuition, notation, and evaluation choices (e.g., WMDP over TOFU for base models), making the paper substantially more rigorous.

**Reviewer JhQa:** We clarified that concepts are defined operationally via the datasets and that our evaluation is deliberately comprehensive, using not just MCQs but also internal probing, activation analysis, and strong adversarial attacks to ensure erasure is robust and deep, not partial. The "spill-over" effect is acknowledged in our limitations.

We believe we have fully addressed all the actionable critiques. Reviewers who maintained their weak reject stance did so with broad final comments. While we are always eager to improve our work, this general feedback did not provide a specific, actionable direction for further revisions.

We respectfully ask the AC to consider the overall reception and the significant and targeted improvements we made in direct response to the reviewers' initial, detailed critiques. We are confident that our submission has been thoroughly validated as a result of the review process.

---

### Decision · Program_Chairs · 2025-09-17

**Decision:**

Accept (poster)

**Comment:**

The paper introduces ELM, a highly novel and principled approach to concept erasure that leverages the model's own introspective classification capabilities. This core idea was praised by multiple reviewers (65Ne, JhQa) as a significant contribution to the field. None of the reviewers identified fundamental flaws with the methodology itself.

The work is supported by a thorough and multifaceted evaluation across multiple models, datasets, and a wide range of metrics. The inclusion of internal knowledge evaluation via probing and activation analysis provides a deeper assessment of erasure that goes beyond simple performance metrics.

During the discussion period, the authors demonstrated commendable responsiveness to reviewer feedback. Crucially, they ran new experiments against three state-of-the-art baselines (RR, TAR, K-FADE), addressing a key concern about insufficient comparisons. These results showed ELM is competitive on erasure while being superior in preserving model fluency—a critical aspect for practical usability. While some technical disagreements with reviewers VH8V and wZRd remain, the authors have diligently addressed the majority of actionable critiques. The positive reviews are well-justified, and the authors' rebuttal has significantly strengthened the submission.

For the camera-ready version, I encourage the authors to better define the boundaries of the concepts they are erasing, as this would further clarify the scope and limitations of their method. Overall, this is a solid paper with a novel contribution and strong empirical support.